# Ensemble Machine Learning Techniques for Accurate and Efficient Detection of Botnet Attacks in Connected Computers

**Stephen Afrifa** [1,2,*], **Vijayakumar Varadarajan** [3,4,5,*], **Peter Appiahene** [2], **Tao Zhang** [1] **and Emmanuel Adjei Domfeh** [2]

[1] Department of Information and Communication Engineering, Tianjin University, Tianjin 300072, China
[2] Department of Computer Science and Informatics, University of Energy and Natural Resources, Sunyani 00233, Ghana
[3] School of Computer Science and Engineering, University of New South Wales, Sydney, NSW 2052, Australia
[4] International Divisions, Ajeenkya D. Y. Patil University, Pune 412105, India
[5] School of Information Technology, Swiss School of Business Management, 1213 Geneva, Switzerland
[*] Correspondence: afrifastephen@tju.edu.cn (S.A.); v.varadarajan@unsw.edu.au (V.V.)

**Abstract:** The transmission of information, ideas, and thoughts requires communication, which is a crucial component of human contact. The utilization of Internet of Things (IoT) devices is a result of the advent of enormous volumes of messages delivered over the internet. The IoT botnet assault, which attempts to perform genuine, lucrative, and effective cybercrimes, is one of the most critical IoT dangers. To identify and prevent botnet assaults on connected computers, this study uses both quantitative and qualitative approaches. This study employs three basic machine learning (ML) techniques—random forest (RF), decision tree (DT), and generalized linear model (GLM)—and a stacking ensemble model to detect botnets in computer network traffic. The results reveled that random forest attained the best performance with a coefficient of determination ($R^2$) of 0.9977, followed by decision tree with an $R^2$ of 0.9882, while GLM was the worst among the basic machine learning models with an $R^2$ of 0.9522. Almost all ML models achieved satisfactory performance, with an $R^2$ above 0.93. Overall, the stacking ensemble model obtained the best performance, with a root mean square error (RMSE) of 0.0084 m, a mean absolute error (MAE) of 0.0641 m, and an $R^2$ of 0.9997. Regarding the stacking ensemble model as compared with the single machine learning models, the $R^2$ of the stacking ensemble machine learning increased by 0.2% compared to the RF, 1.15% compared to the DT, and 3.75% compared to the GLM, while RMSE decreased by approximately 0.15% compared to the GLM, DT, and RF single machine learning techniques. Furthermore, this paper suggests best practices for preventing botnet attacks. Businesses should make major investments to combat botnets. This work contributes to knowledge by presenting a novel method for detecting botnet assaults using an artificial-intelligence-powered solution with real-time behavioral analysis. This study can assist companies, organizations, and government bodies in making informed decisions for a safer network that will increase productivity.

**Keywords:** machine learning; botnet; malware; network traffic; ensemble model; IoT

## 1. Introduction

Information exchange between two or more people, or between equipment, is referred to as communication [1]. The transmission of information, ideas, and thoughts requires communication, which is a crucial component of human contact. With the advent of technology, information is currently exchanged easily, and its influence is enormous [2]. For instance, computers are networked together to communicate. A computer network is a collection of linked computers and other devices that can interact with one another [3]. Typically, wired or wireless technology is used to create this connection [4]. The utilization of Internet of Things (IoT) devices is a result of the advent of enormous volumes of messages delivered over the internet. However, as Internet of Things (IoT) devices proliferate, the

number of IoT-based assaults has been steadily increasing [5]. The IoT botnet assault, which attempts to perform genuine, lucrative, and effective cybercrimes, is one of the most important IoT dangers. A network of personal computers that have malicious software installed on them and are managed collectively without the owners' knowledge is known as a "botnet" [6,7]. These infected machines, commonly referred to as "bots", are capable of carrying out a number of operations, such as sending spam emails, starting distributed denial-of-service (DDoS) assaults, and collecting personal data. The academic community has paid close attention to the recent increase in botnet activity in cyberspace. Because they only consume a small amount of processing resources, botnets are difficult to detect [8]. The internet may not be threatened by a single bot, but a network of bots can surely cause a lot of damage [9]. Currently, botnets are the primary malware platform threat that hackers utilize, and they are the source of the majority of internet security problems [10,11]. The methods used to transmit malware are evolving in new and creative ways. The malware is then employed to conduct further attacks, such as data exfiltration and denial of service (DoS) assaults, either using or on infected machines. According to Stevanovic et al. [12], 40% of computers around the globe are part of botnets. This necessitates quick action to detect botnets and provide safer online communication. Early identification of botnet attacks lessens the damage brought on by potential attacks. Over the last few decades, academicians have suggested and implemented a variety of strategies aimed at improving botnet detection [13,14]. The artificial intelligence (AI) and machine learning (ML) techniques are among the methods for detecting botnets. IoT devices and other edge systems can benefit from threat mitigation using machine learning (ML) techniques with a reasonable amount of accuracy [15]. The collection of characteristics used for the categorization of harmful activity determines how well AI and ML approaches function.

In this study, botnets are detected in computer network traffic data far more accurately using machine learning and ensemble approaches. To create one ideal prediction model, the ensemble method is used [16]. Additionally, the recent research can determine if an incoming packet of activity is coming from a bot or not. Furthermore, the study offers benchmark practices to aid organizations and agencies in preventing and/or mitigating botnet attacks using a qualitative approach. The current study also provides a novel approach to the topic of botnets in the literature. The study's quantitative and qualitative approaches will help in the identification and prevention of botnet attacks, increasing efficiency and production.

The following are the primary contributions of this study, as adapted from [17,18] in a study:

(1) To identify botnet attacks on connected computers, a novel technique based on single machine learning and stacking ensemble models is presented.
(2) An artificial-intelligence-powered system for detecting botnet attacks on connected computers and preventing botnet activity in real time is presented.
(3) The study uses both qualitative and quantitative approaches to give several perspectives on the research topic of preventing and detecting botnet attacks on connected computers.
(4) This study proposes a data-driven strategy to botnet attack prevention for governments, agencies, and organizations.
(5) It is a novel contribution to the literature in which a new model for detecting botnet attacks is proposed.

The remainder of the research is structured as follows: Section 2 discusses related material that takes into account machine learning strategies for botnet detection. The materials and procedures used in the investigation are presented in Section 3. It outlines the study's conceptual framework and the statistical methods used to develop the models. The experimental findings of the various models used in the proposed framework are presented in Section 4. The section "Discussion", Section 5, provides an in-depth analysis of the study's findings. The study's conclusion and future works are presented in Section 6.

## 2. Related Works

Numerous methods have been created to automatically recognize and often classify communication streams. The study is buttressed with related literature that utilized machine learning approaches. To begin with, a methodology for the preprocessing of the IoT bot dataset and categorization of the various attack types were described by Motylinski et al. [19]. They compared the outcomes of random forest, k-nearest neighbor (kNN), support vector machine (SVM), and logistic regression (LR) classifiers that were graphics processing unit (GPU) accelerated, as well as the preprocessing procedures used to prepare the data for training. The training and estimation durations were greatly shortened by using feature selection and training models on GPU. It must be emphasized that their study did not provide measures to fight botnets qualitatively. Additionally, Akash et al. [20] created a model that takes into consideration the botnet identification using machine learning (ML) methods. Their algorithm looked for botnet-like irregularities in a group of IoT devices that were seeking to connect to a network. However, the study only took into account two ML algorithms, while considering a model to identify botnets.

In another related study by Asadi [21], long short-term memory (LSTM), autoencoder, and support vector machine (SVM) are three techniques that were used with cooperative game theory to identify IoT botnet assaults. When compared to earlier research, their suggested technique enhanced accuracy by 11.624%, recall by 11.629%, and learning time for SVM by 154.41 s. However, the study did not take into account the suggested model's temporal complexity to determine the model's rate of detection. To locate the automated bot accounts in the Twitter network using both fundamental and derived features, Gera and Sinha [22] employed the T-Bot bot identification framework, which is AI-driven. To attain optimality in determining the automation score to the suspected bots in trend-centric social networks, the parameters of machine learning models were fine-tuned. Utilizing a novel centroid initialization technique (CIA), the suggested T-Bot made it easier to detect bots in trend-centric datasets, especially when dealing with imbalanced datasets. To improve the security of IoT enabled networks used for network traffic of smart cities, Onyema et al. [23] presented an ensemble intrusion approach based on cyborg intelligence (machine learning and biological intelligence) architecture. Using the KDDcup99 dataset, many algorithms, including random forest, Bayesian network (BN), C5.0, CART, and artificial neural network, were examined to see how well they would recognize threats and attacks/botnets in IoT networks based on cyborg intelligence. Their findings showed that the AdaBoost ensemble learning based on the cyborg intelligence intrusion detection framework allowed for the facilitation of numerous network features with the potential to quickly recognize various botnet attacks.

In a separate study, Okey et al. [24] suggested a technique for identifying network intrusions and assaults based on boosted machine learning (ML) classifiers. According to experimental findings, their suggested model performed better than current ensemble models in terms of assessment criteria. According to their experimental findings, BoostedEnsML performed better in terms of accuracy than already-existing ensemble models. Furthermore, Alrayes et al. [25] developed a botnet detection model for the IoT context utilizing the barnacles mating optimizer with machine learning (BND-BMOML). The BND-BMOML model that was presented was focused on applying machine learning classifiers to identify and recognize botnets in the IoT context. A benchmark dataset was used to experimentally validate the BND-BMOML approach, and the results showed notable performance gains over other studies. To fight volumetric DDoS (VMFCVD) assaults, Prasad and Chandra [26] suggested a voting-based multimode system. Their proposed model outperformed current existing studies based on their experiments. They found that, during their trial, fast detection mode (FDM) was able to reduce dimensions by more than 97% while generally maintaining accuracy levels of 99.9%. When the server was the target of a DDoS assault, the VMFCVD worked remarkably well.

The k-nearest neighbor (kNN) technique was applied to categorize botnet assaults in the IoT context in research by Syamsuddin and Barukab [27]. They used a variety of

feature selection strategies to train their dataset, which increased the kNN's effectiveness in categorizing IoT botnet assaults. The authors modified the kNN algorithm to create a hybrid model that performed best among competitors, having the highest level of accuracy and quickest execution time. Finally, Yang et al. [28] introduced a unique parallel detection model called N-Trans based on the N-gram method with the Transformer model that can discriminate between trustworthy and malicious domain names with accuracy. According to their experimental findings, the parallel detection model based on N-gram and Transformer is 96.97% accurate in detecting harmful domain names in the domain generation algorithm (DGA).

## 3. Methods

This section presents the framework employed for this study and the processes involved to train the dataset. Additionally, the evaluation metrics utilized are explained.

### 3.1. Data Collection

Adequately labeled dataset for botnet detection is rare to find. This study is based on a publicly available dataset from the FigShare data repository via the link (https://doi.org/10.6084/m9.figshare.21769658.v1) (accessed on 10 December 2022). The dataset is already labeled and used for botnet detection analysis. This study uses these data as a benchmark dataset. This traffic is based on real-world situations and devices that employ Internet Protocol (IP) addresses as well as the frequency with which botnets connect to a specific computer. The computers are connected to one another.

### 3.2. Data Preprocessing

The dataset was preprocessed to convert raw data into valuable information for the machine learning techniques to understand [29]. It is believed that the data are mostly incomplete and contain a lot of errors. The steps involved in the preprocessing stage are handling of null or missing values, label encoding, and feature selection. Handling of null or missing values deals with checking if there are missing values, however, the dataset had no missing or null values and/or no redundant data. Additionally, label encoding was performed to transform the data to feed the models. One of the most crucial steps of preprocessing is the feature selection [14]. The feature selection process was employed to select the most suitable features to extract meaningful data during the statistical technique.

### 3.3. The Proposed Framework

The study followed the steps to carry out a machine learning project where the dataset was collected from the FigShare data repository and passed through the proposed machine learning classifiers. The proposed framework of the study is represented in Figure 1. The dataset was trained using the proposed machine learning models, which include the random forest, decision tree, generalized linear model, and stacking ensemble of models. The result depicts either legitimate or botnet traffic. This provides a solution powered by artificial intelligence that can detect it in real-time behavioral analysis, block it, and stop any botnet activity. To determine the robustness of the models, the performance of the given model is evaluated using evaluation metrics.

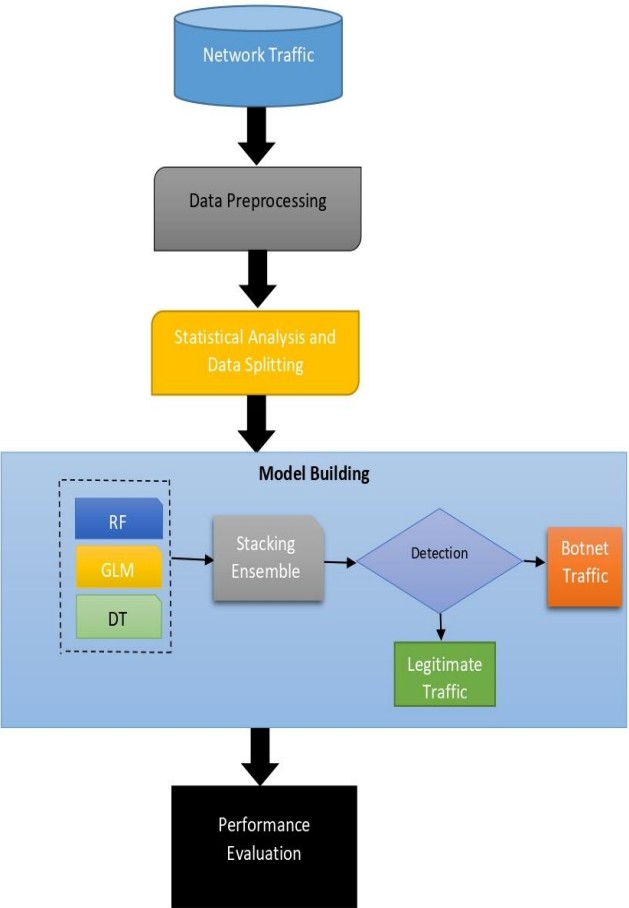

**Figure 1.** The modeling framework of the study.

### 3.4. Statistical Analysis and Data Splitting

The dataset was passed through some statistical techniques to obtain meaningful insights into the attributes of the dataset. Exploring the data to obtain meaning insights help the training of the models. Furthermore, the data were divided into training and testing sets with 70% and 30% division, respectively. The testing dataset helps to evaluate the performance of the models [30]. The training and testing dataset components were used to train the proposed models.

### 3.5. The Machine Learning Classifiers

This study employs three single machine learning models, the random forest (RF), generalized linear model (GLM), and decision trees (DT). The three single machine learning models were used to build a stacking ensemble model to also train the dataset. The non-parametric supervised learning approach used for classification and regression applications is the decision tree [31]. By using a greedy search to find the ideal split points inside a tree, decision tree learning uses a divide and conquer technique [32]. Once most or all records have been categorized under certain class labels, this dividing procedure is then repeated in a top-down, recursive fashion [30]. The random forest (RF) creates decision trees by segmenting samples and seeks the optimal outcome in accordance with the voting prediction; it is difficult to overfit while addressing regression issues [16]. Additionally, even if the link between the predictors and the answer is not linear, generalized linear model (GLM) models enable us to construct a linear relationship between the two [33]. A link function, which connects the response variable to a linear model, enables this. An individual machine learning model was layered on top of a higher-order model in stacking, a type of meta-learning or ensemble learning [34]. The fundamental principle of a stacking ensemble is to employ level-0 basic learners to create metadata or input, and level-1 meta

learners to analyze that metadata [7]. According to earlier studies, the stacking model increases the simulation's accuracy. In this study, the generalized linear model (GLM) was chosen as the meta learner, whereas RF, DT, and GLM were utilized as the basic learners. Researchers may discover which algorithms produce the best/better predictions from datasets with the use of meta-learning [35]. Due to the stochastic, systematic, and link function components the GLM contains, it is chosen as the meta learner. Regression analysis and variance analysis for multiple dependent variables by one or more component variables or covariates are both provided by the GLM procedure [36], which is congruent with the regression dataset utilized in this work. An ensemble is made up of a group of learners known as base learners [37]. As base learners, the random forest (RF), decision tree (DT), and generalized linear model (GLM) were utilized to focus on accurately identifying the most highly weighted samples while avoiding over-fitting during model training [17]. The three machine learning algorithms were used to generate a set of hypotheses and combine them to achieve the best outcomes [38]. The developed models are able to detect whether a network is legitimate or detect botnet traffic to help in making informed decisions for companies, agencies, and organizations.

### 3.6. Performance Evaluation

The performance of the machine learning models was assessed in this study using the root mean squared error (RMSE), mean absolute error (MAE), mean absolute percentage error (MAPE), and coefficient of determination ($R^2$). To fully depict the error distribution, the RMSE or the standard error (SE) were used [39]. The performance of the ML models improves when RMSE and MAE decrease and $R^2$ approaches 1 [39,40]. MAPE is a different statistical metric for assessing the accuracy of a regression model in terms of differences between observed and predicted values. A lower MAPE denotes a high prediction model accuracy rate [41]. When computing statistical indicators, it is assumed that simulations and observations of varying durations have the same weight. Equations (1)–(4) define the aforementioned evaluation models as follows:

$$RMSE = \sqrt{\sum_{i=1}^{n} \frac{\left( y_{observed} - \hat{y}_{predicted} \right)^2}{n}} \tag{1}$$

$$MAE = \frac{1}{n} \sum_{i=1}^{n} \left| y_{observed} - \hat{y}_{predicted} \right| \tag{2}$$

$$MAPE = \frac{100\%}{n} \sum_{k=1}^{n} \left| \frac{y_{observed} - \hat{y}_{predicted}}{y_{observed}} \right| \tag{3}$$

$$R^2 = \left( \frac{\sum_{i=1}^{n} \left( y_{observed} - y_{average} \right) \left( \hat{y}_{predicted} - \hat{y}_{average} \right)}{\sqrt{\sum_{i=1}^{n} \left( y_{observed} - y_{average} \right)^2 \sum_{i=1}^{n} \left( \hat{y}_{predicted} - \hat{y}_{average} \right)^2}} \right)^2 \tag{4}$$

where $y_{observed}$ represents the observed values, $\hat{y}_{predicted}$ is the predicted values, $n$ is the number of samples, and $y_{average}$ and $\hat{y}_{average}$ represent the average observed and predicted values, respectively.

## 4. Experimental Results

This section presents the results obtained from the developed model and the performance of the machine learning classifiers.

### 4.1. Statistical Analysis of the Dataset

The dataset utilized for this study encompasses the computer network traffic with the local Internet Protocol (IP) address. IP addresses enable computing devices to communicate

with destinations such as websites and streaming services, as well as inform websites about who is connecting. The total number of IP addresses is 10, ranging from 0–9. The statistical analysis of the data helps to ascertain the flow of the IP within the week. The days are categorized from 0–6, thus Sunday–Saturday. The flow of the IP within the week is represented in Figure 2.

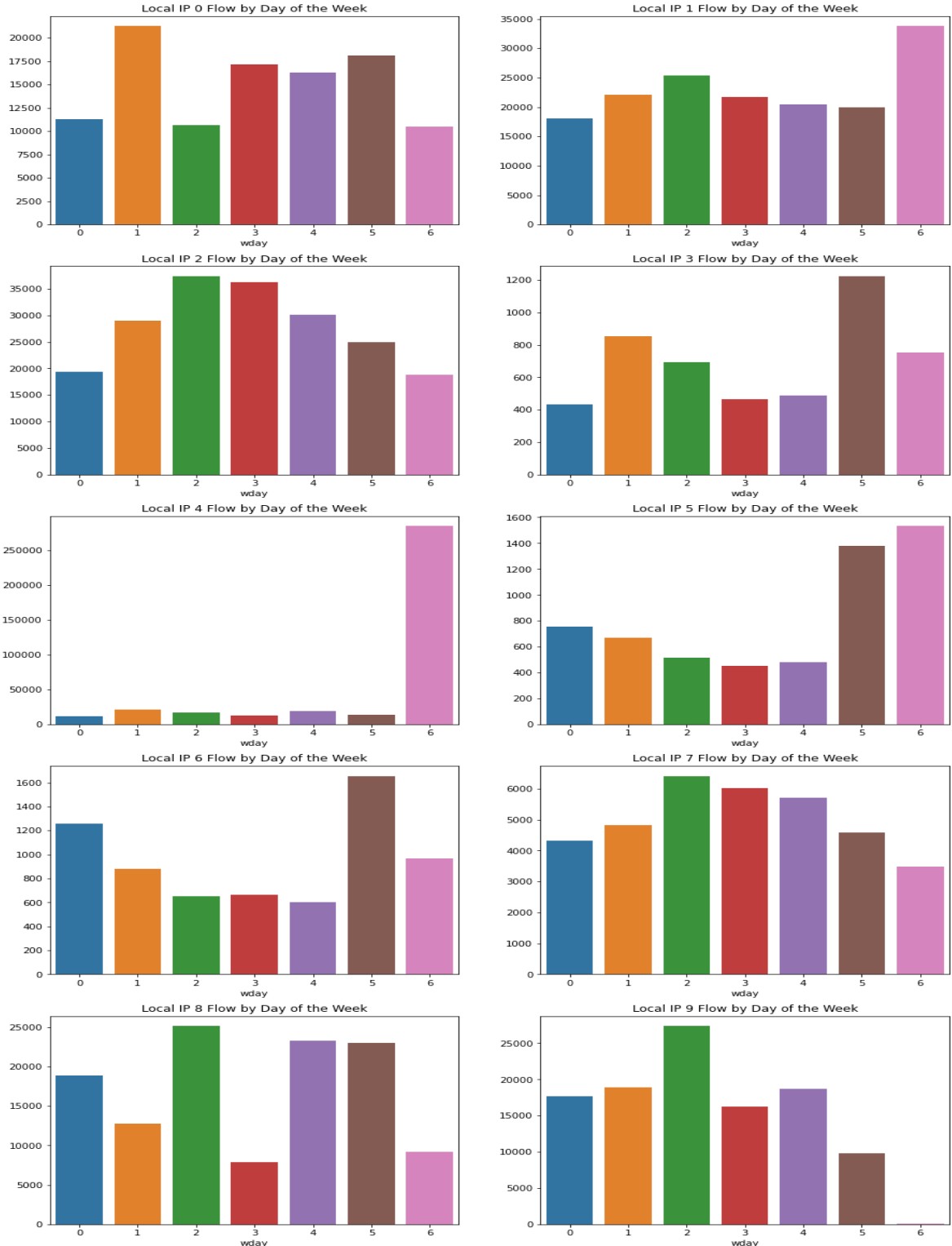

**Figure 2.** The flow of IP by day of the week.

Additionally, the flow reaches its peak in each of the 10 IP addresses in the connected network traffic. The packets that come in IP 0 reach their peak at 5990. Figure 3 represents the flow of local IP addresses and the maximum packets they achieve.

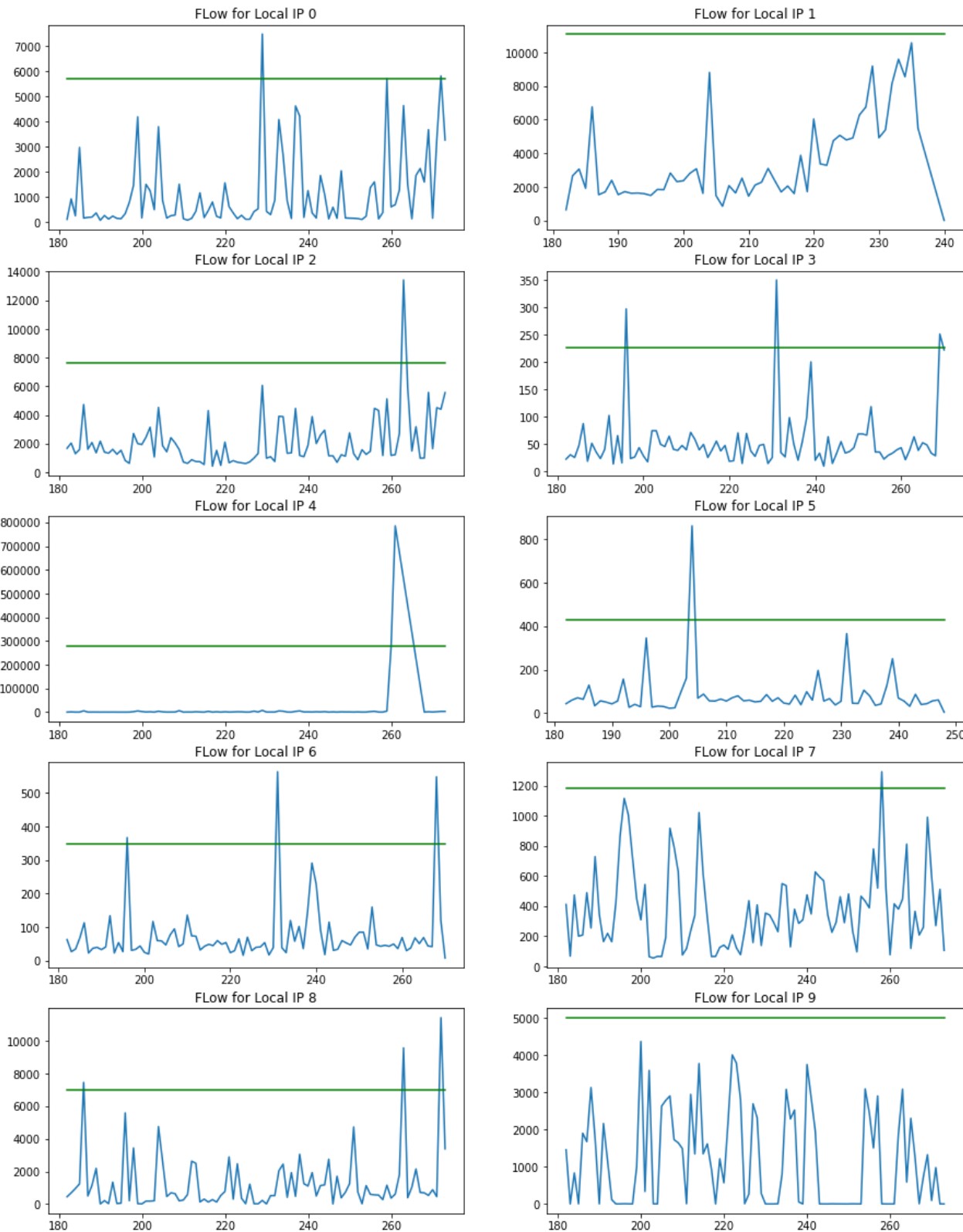

**Figure 3.** The flow of the IP addresses and maximum packets.

## 4.2. Performance of the Stacking Ensemble and Basic Machine Learning Models

Based on the connected network traffic data, we verified the performance of three basic machine learning (ML) models and the stacking ensemble ML model. The dataset was randomly divided into a training set and a test set according to 70% and 30%, a proportion commonly used in ML [42].

Table 1 shows the performance of applying various single machine learning and stacking ensemble machine learning models in the proposed framework. There was a little difference in model performance among the basic and stacking ensemble models. The random forest attained the best accuracy ($R^2$ of 0.9977), followed by decision tree with an $R^2$ of 0.9882, while GLM was the worst among the basic machine learning models with an $R^2$ of 0.9522. It can be observed that the random forest and decision tree performed closely well due to the similarities of the model behavior. The decision trees are very easy as compared to the random forest. A decision tree combines some decisions, whereas a random forest combines several decision trees [42]. Almost all ML models have achieved satisfactory performance, with an $R^2$ above 0.93. Overall, the stacking ensemble model obtained the best performance, with an RMSE of 0.0084 m, MAE of 0.0641 m, $R^2$ of 0.9997.

**Table 1.** Evaluation of model performances.

| Model | Evaluation | Values |
|---|---|---|
| GLM | $R^2$ | 0.9522 |
| | MAE | 0.0852 |
| | RMSE | 0.0099 |
| | MAPE | 0.9952 |
| DT | $R^2$ | 0.9882 |
| | MAE | 0.0752 |
| | RMSE | 0.0085 |
| | MAPE | 0.9853 |
| RF | $R^2$ | 0.9977 |
| | MAE | 0.0715 |
| | RMSE | 0.0099 |
| | MAPE | 0.0952 |
| Stacking Ensemble | $R^2$ | 0.9997 |
| | MAE | 0.0641 |
| | RMSE | 0.0084 |
| | MAPE | 0.0899 |

In contrast, the GLM performed the worst, with an RMSE of 0.0099 m, MAE of 0.0852 m, $R^2$ of 0.9522. Compared with the basic models, the $R^2$ of the stacking ensemble machine learning increased by 0.2% compared to the RF, 1.15% compared to the DT, and 3.75% compared to the GLM, while RMSE decreased by approximately 0.15% compared to the GLM, DT, and RF single machine learning techniques. Based on the experimental results, the stacking ensemble model achieved the highest accuracy when compared with the single machine learning models, with an $R^2$ of 0.9997. This study found that the stacking ensemble model is better to use in the detection of legitimate or botnet computer traffic. The models produced substantial results and a novel strategy that surpasses those in studies by Duan et al. [43] and Akash et al. [20]. Based on the results, the current study includes the most recent enhanced ensemble model for detecting botnet attacks on connected computers.

## 4.3. Time Complexity of the Machine Learning Models

It should be noted that computational complexity is still important in the study of large systems. Several tools have been developed in recent years to assist in reducing the complexity of workflows and handling interactions with high performance computing resources for efficiency and productivity [44]. For the single machine learning models, the time complexity was 45 s for the random forest, 50 s for the decision tree, and 55 s for the generalized linear model. The stacking ensemble machine learning process took 2 min

and 45 s to complete and had the best prediction, however, it had the worst time. The stacking ensemble's long execution time was due to the process involved; thus, the art of ensembling is difficult to learn, and any incorrect selection can lead to lower predictive accuracy compared to an individual model [16]. The time complexity of the machine learning classifiers is summarized in Table 2.

**Table 2.** Time complexity of the machine learning models.

| Algorithm | Parameter | Time Complexity (s) |
|---|---|---|
| Random forest | Best training time | 0.45 |
| Decision tree | Best prediction | 0.50 |
| Generalized linear model | Worst training time | 0.55 |
| Stacking ensemble | Best prediction and worst training time | 2.45 |

*4.4. Benchmark Practices to Prevent Botnet Attacks*

Computer network communication is an integral part of every organization in today's growing technological framework. It is empirical to also practice safe and efficient practices to prevent botnets. The effects of botnets are enormous and not easily determined. This calls for immediate and stringent measures by companies, organizations, and agencies to train their staff and/or employees to mitigate the effects of botnets. The DDoS assaults are among the worst and hardest to contain of the many distinct cybersecurity dangers that occur online. A botnet is renowned for its ability to spread infection to other devices it interacts with after compromising a computer, for instance by automatically sending spam emails. This study presents some empirical performances qualitatively to mitigate the effects of botnets.

To begin with, in order to prevent botnet assaults, it is crucial to make sure your entire system is up to speed with the latest viruses and malware. Botnet assaults make use of application and software weaknesses; hence it is advised that organization information technology (IT) specialists update their systems on a regular basis to keep current. Additionally, company IT experts should keep a close eye out for any strange activity on your network. If a company has a better grasp of your normal traffic and how everything typically performs, this will be considerably more successful. The IT experts can employ a time monitoring scheme, such as 12/24 h monitoring network exercises. Hackers attempt to input various login credentials on networks in order to access the system; as a result, all unsuccessful login attempts on a network should be tracked as they occur. It is also possible that hackers are attempting to connect into the network using stolen login information. Additionally, this necessitates routine password changes for network users and workers. There is powerful software available now that combats botnets for a price. Businesses are recommended to make significant investments to combat botnets. Before any botnet activity reaches your web server, an artificial-intelligence-powered solution may detect it in real-time behavioral analysis, block it, and stop any botnet activity. Information and data are firm resources that cannot be quantified; thus, it is important to secure them in order to avoid any unanticipated events.

## 5. Discussion

With the advancement of technology, information is now easily transmitted, and its impact is enormous. Good information is required for efficient corporate operations and decision making at all levels. Interconnected computers promise a bright future for the brilliant and efficient exchange of information and resources, increasing productivity in many sectors of the economy. Interconnection provides low-latency, high-availability links that allow businesses to move data across these assets with confidence. Botnet attacks have become a hazard and danger to network and internet security in recent years. They include a number of harmful operations in network traffic. Botnets are made up of independent robot and network components. The botmaster programs and builds the bot for certain

objectives, utilizing computers termed as zombies. Botnets are incredibly widespread and may influence millions of computers. These bots are managed by one or more attackers known as botmasters, with the goal of carrying out harmful acts. It is important to develop a way to avoid botnet attacks in networked computers for safer and more efficient operations and resource sharing. Using both quantitative and qualitative methodologies, the study proposed a strategy to prevent botnet attacks on connected computers. According to the findings, the flow of IP reaches its peak in each of the ten (10) IP addresses in the connected network traffic.

To detect genuine or botnet network activity on connected computers, the researchers utilized three single machine learning models and an ensemble model. The random forest, decision tree, and generalized linear model are the single machine learning methods used in this study. Additionally, the stacking ensemble model was employed to also train the dataset to determine botnet attacks. Among the single machine learning models, the random forest performed better with an accuracy ($R^2$) of 0.9977, followed by decision tree with 0.9882. The generalized linear model achieved an accuracy of 0.9522, which represents the least performed model among the single machine learning models. Furthermore, the stacking ensemble model used the generalized linear model as a meta learner, as well as the random forest, decision tree, and generalized linear model as base learners. It is believed that the stacking ensemble model increases model accuracy and performance, according to research. Based on the experimental results, the stacking ensemble model achieved the highest accuracy when compared with the single machine learning models, with an $R^2$ of 0.9997. This study found that the stacking ensemble model is better to be used in the detection of legitimate or botnet computer traffic. The coefficient of determination ($R^2$) of the stacking ensemble machine learning increased by 0.2% compared to the random forest (RF), 1.15% compared to the decision tree (DT), and 3.75% compared to the generalized linear model (GLM), while root mean square error (RMSE) decreased by approximately 0.15% compared to the GLM, DT, and RF single machine learning techniques. This study illustrated the impact of ensemble and single machine learning models to detect botnets in connected computers. Furthermore, the study also revealed that it is necessary to continuously monitor the operations of connected computer networks for easy and better communication. Computer networks are essential to contemporary civilization because they allow for the transmission and sharing of information between people, groups, and objects. Computer networks have significantly influenced society by facilitating worldwide connection and information sharing, therefore, businesses are recommended to make significant investments to combat botnets. Before any botnet activity reaches your web server, an artificial-intelligence-powered solution may detect it in real-time behavioral analysis, block it, and stop any botnet activity. Information and data are firm resources that cannot be quantified; thus, it is important to secure them in order to avoid any unanticipated events. Table 3 compares the obtained findings to those of previously existing models to illustrate the proposed model's dependability and robustness.

**Table 3.** Comparison of the obtained results with existing published studies.

| Published Papers | Year of Publication | Method and Results of the Published Paper | Performance of the Proposed Model |
|---|---|---|---|
| Disha and Waheed [38] | 2022 | Decision tree (90.15%) | |
| Rehman Javed et al. [7] | 2022 | Decision tree (97.8%) | Decision tree (98.82%) |
| Okey et al. [24] | 2022 | Decision tree (98.7%) | |
| Okey et al. [24] | 2022 | Random forest (98.4%) | |
| Onyema et al. [23] | 2022 | Random forest (99.0%) | Random forest (99.77%) |
| Hosseini et al. [15] | 2022 | Random forest (97.0%) | |
| Vimont et al. [36] | 2022 | Generalized linear model (81.9%) | Generalized linear model (95.22%) |
| Alhogail and Al-Turaiki [45] | 2022 | Generalized linear model (91.66%) | |
| Xu et al. [46] | 2022 | Generalized linear model (90.5%) | |
| Akhtar and Feng [47] | 2022 | Stacking ensemble (99.0%) | Stacking ensemble (99.99%) |
| Masoudi-Sobhanzadeh and Emami-Moghaddam [14] | 2022 | Stacking ensemble (90.0%) | |
| Yerima and Bashar [8] | 2022 | Stacking ensemble (96.0%) | |

The proposed model surpassed all relevant articles published in terms of performance, as shown in Table 3 above.

## 6. Conclusions and Future Works

Computer networks have significantly influenced society by facilitating worldwide connection and information sharing. The use of Internet of Things devices has influenced communication, thus, digital communication, enormously. However, the effects and attacks on IoT devices are worrying. In IoT contexts, botnet-based cyberattacks are multi-staged assaults that often involve scanning activities and distributed denial of service. Researchers have proposed several techniques to combat botnets, however, they are limited on the specific dataset used. The study uses both qualitative and quantitative approaches to give several perspectives on the research topic of preventing and detecting botnet attacks on connected computers. This current study provides novel approaches to detect botnets using three single machine learning and stacking ensemble machine learning techniques to detect botnet attacks in connected computers. Regarding the stacking ensemble machine learning compared with the single machine learning models, the coefficient of determination ($R^2$) of the stacking ensemble machine learning increased by 0.2% compared to the random forest, 1.15% compared to the decision tree, and 3.75% compared to the generalized linear model, while root mean square error decreased by approximately 0.15% compared to the generalized linear model, decision tree, and random forest single machine learning techniques. This study found that the stacking ensemble model is better to use in the detection of legitimate or botnet computer traffic. Additionally, the study revealed that companies employ botnet prevention techniques to increase network efficiency. Businesses are recommended to make significant investments to combat botnets. Before any botnet activity reaches your web server, an artificial-intelligence-powered solution may detect it in real-time behavioral analysis, block it, and stop any botnet activity. Computer networks are essential to contemporary civilization because they allow for the transmission and sharing of information between people, groups, and objects. Computer networks have significantly influenced society by facilitating worldwide connection and information sharing, therefore, businesses are recommended to make significant investments to combat botnets. Future training of the model with a huge data collection is possible. Furthermore, machine learning classifiers such as support vector machine, artificial neural network, and k-nearest neighbor can be used. Therefore, it is necessary to continuously obtain a larger network traffic dataset to attain higher accuracy in the future.

**Author Contributions:** Conceptualization, V.V., T.Z., and P.A.; methodology, S.A. and P.A.; software, S.A. and E.A.D.; validation, V.V., T.Z., and P.A.; formal analysis, P.A. and T.Z.; investigation, V.V. and T.Z.; resources, V.V.; data curation, S.A. and E.A.D.; writing—original draft preparation, S.A.; writing—review and editing, P.A., V.V., and T.Z.; visualization, S.A. and E.A.D.; supervision, T.Z., V.V., and P.A.; project administration, V.V. and T.Z. All authors have read and agreed to the published version of the manuscript.

**Funding:** This research received no external funding.

**Institutional Review Board Statement:** Not applicable.

**Informed Consent Statement:** Not applicable.

**Data Availability Statement:** The data used for the study are available in the FigShare data repository via Afrifa, Stephen (2022): Botnet Detection. FigShare. Dataset. https://doi.org/10.6084/m9.figshare.21769658.v1.

**Acknowledgments:** The authors are grateful to Adwoa Afriyie for her encouragement and advice throughout the study.

**Conflicts of Interest:** The authors declare no conflict of interest.

## Abbreviations

Definition of variables, sets, notations, and symbols used in the study.

| | |
|---|---|
| RF | Random forest |
| DT | Decision tree |
| GLM | Generalized linear model |
| $R^2$ | Coefficient of determination |
| @ | Location or institution of an email recipient |
| \n | Newline |
| # | Pound sign used as prefix for an address |
| IoT | Internet of Things |
| IP | Internet Protocol |
| $\hat{y}_{average}$ | Average predicted |
| $y_{average}$ | Average observed |
| $\hat{y}_{predicted}$ | Predicted values |
| ipn | Interpenetrating network |
| rpn | Region proposal network |
| f | Frequency of connected botnet computers |

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
