# Peer review of "Ensemble Machine Learning Techniques for Accurate and Efficient Detection of Botnet Attacks in Connected Computers"

_2673-4117, doi:10.3390/eng4010039_

Round 1
Reviewer 1 Report
In this paper, the authors present the idea of efficient detection of botnet attacks on connected computers through machine learning techniques. For this purpose, authors use a machine learning techniques like random forest. decision tree, and generalize linear model, and a stacking ensemble model. The ensemble model is reported to produce the best results.
The idea is good and the results are fine too. However, I have a few comments on the quality of the paper.
1. There are several mistakes in the paper. For example on line 78, there is a bizarre sentence that does not make sense to me.
2. In section 2 of the paper, the authors review the related work. However, the the whole section seems like some information report and it lacks any critical analysis of the related work. The authors need to improve on this.
3. In sections 3.3, authors explain that they use GLM as meta learner and use RF, DT, and GLM as basic learners. However, authors do not provide any basis for this decision. More in-depth discussion is required on this part.
4. Experimental results section number is incorrect.
5. More discussion is required in section 5. The discussion presented seem superficial.
6. Abbreviations in conclusion section seem a bit out of place.
Author Response
To begin with, we are grateful for your kind and expert review of our article. All the comments raised have helped to improve our study and have enriched our work. The comments and action taken have been highlighted in the manuscript for your perusal. The authors have considered and addressed the comments wholeheartedly to the best of our knowledge. Literature contributes to knowledge, and we are by this work contributing our quota to the academic space. There are more room for improvement, which could also be addressed in future studies. The authors have also contacted a professional English colleague to proofread the whole manuscript for any grammatical problems and/or errors. We believe our action taken would meet your kindest expectations.
Thank you very much for your kindest consideration and acceptance.

Reviewer 2 Report
The submitted paper presents “Ensemble Machine Learning Techniques for Accurate and Efficient Detection of Botnet Attacks in Connected Computers". The authors have made a good effort for this review. However there are lot of problems in present form of manuscript. There are a number of language problems and theoretical deficiencies in this paper as.
Language problems:
1. A thorough revision of English language is required for whole paper.
2. Page 2: line 61 the word academics should be replaced by academicians
3. The section headings like Materials and Methods and Analyzing don’t convey the standard meanings
Theoretical deficiencies:
1. The abstract of the paper should thoroughly revised it doesn’t highlight the real objective of this paper. It is harder to identify the contributions of this paper after reading abstract of the paper.
2. The Introduction section doesn’t highlight the contributions of this paper.
3. The experimental Results are very superficial don’t indicate any in-depth analysis of the problem.
4. Analysis section is very irrelevant doesn’t explain the real problem formulated in the paper, the authors are unable to explain the results obtained, they just rely on machine learning output results, which is very unrealistic. Similarly reasoning is applicable to Conclusion and Future Works.
Author Response
The authors are grateful for your kind and expert review of our study. All the suggestions and comments raised have enriched and improve our study. The comments and actions taken have been highlighted in the manuscript for your perusal. The authors have considered and addressed the comments wholeheartedly to the best of our knowledge. Literature contributes to knowledge, and we are by this work contributing our quota to the academic space. There are more room for improvement, which could also be addressed in future studies. The authors have also contacted a professional English colleague to proofread the whole manuscript for any grammatical problems and/or errors. We believe our action taken would meet your kindest expectations.
Thank you very much for your kindest consideration and acceptance.

Reviewer 3 Report
1. The ensemble learning algorithm is proposed in this study based on some basic algorithms. However, why to select and use them? The motivation of this paper is not well stated in the introduction.2. The justification of selection the machine learning algorithm is not clear. for example why using DT and RF?
3. There are several abbreviations that are neither defined nor explained.
4. Experimental results are week.
5. Related work needs to be clearly structured. Authors did not make any real effort to synthesize the major existing contributions and to put them into clear categories based on the different approaches. Also, the manuscript could be substantially improved by relying and citing more on recent literature.
6. Having a table that summarizes the variables, sets, and notations will facilitate the reading of the paper. Please add a table with all the variables and sets used in the system model.
7. The discussion section in the present form is relatively weak and should be strengthened with more details and justifications.
8.The authors are not making real effort in producing the figures. All figures are not represented and bad resolution. Also, some figures without x-axis and y-axis labels.9.What is the computational complexity of the proposed solution?
10.The dataset description and the experimental procedures not explained.
Author Response
Thank you very much for your kind and expert review of our article. The suggestions, comments, and corrections made have improved our study and we appreciate them. The comments and action taken have been highlighted in the manuscript for your perusal. The authors have considered and addressed the comments wholeheartedly to the best of our knowledge. Literature contributes to knowledge, and we are by this work contributing our quota to the academic space. There are more room for improvement, which could also be addressed in future studies. The authors have also contacted a professional English colleague to proofread the whole manuscript for any grammatical problems and/or errors. We believe our action taken would meet your kindest expectations.
Thank you very much for your kindest consideration and acceptance.

Round 2
Reviewer 1 Report
Authors have addressed most of my comments. So, the paper can be accepted in its current form.
Author Response
The authors are grateful for your time to review our paper. Your effort and comments really shaped our manuscript and appreciate your time.
Reviewer 3 Report
several points still without clear answer (amendment) such as:
6. Having a table that summarizes the variables, sets, and notations will facilitate the reading of the paper. Please add a table with all the variables and sets used in the system model.
7. The discussion section in the present form is relatively weak and should be strengthened with more details and justifications.
8.The authors are not making real effort in producing the figures. All figures are not represented and bad resolution. Also, some figures without x-axis and y-axis labels.
9.What is the computational complexity of the proposed solution?
10.The dataset description and the experimental procedures not explained.
Author Response
Thank you very much for your kind and expert review of our article. The suggestions, comments, and corrections made have improved our study and we appreciate them. The comments and action taken have been highlighted in the manuscript for your perusal. The authors have considered and addressed the comments wholeheartedly to the best of our knowledge. Literature contributes to knowledge, and we are by this work contributing our quota to the academic space. There are more room for improvement, which could also be addressed in future studies. The authors have also contacted a professional English colleague to proofread the whole manuscript for any grammatical problems and/or errors. We believe our action taken would meet your kindest expectations.

Round 3
Reviewer 3 Report
The authors have completed all amendments.